# Treasures of Italian Microbial Culture Collections: An Overview of Preserved Biological Resources, Offered Services and Know-How, and Management

Marino Moretti [1,†], Jacopo Tartaglia [1,†], Gian Paolo Accotto [2], Maria Serena Beato [3], Valentina Bernini [4], Annamaria Bevivino [5], Maria Beatrice Boniotti [6], Marilena Budroni [7], Pietro Buzzini [8], Stefania Carrara [9], Federica Cerino [10], Clementina Elvezia Cocuzza [11], Roberta Comunian [12], Sofia Cosentino [13], Antonio d'Acierno [14], Paola De Dea [15], Laura Garzoli [16], Maria Gullo [17], Silvia Lampis [18], Antonio Moretti [19], Alda Natale [20], Giancarlo Perrone [19], Anna Maria Persiani [21], Iolanda Perugini [1], Monica Pitti [22], Annarita Poli [23], Antonino Pollio [24], Anna Reale [14], Annamaria Ricciardi [25], Cristiana Sbrana [26], Laura Selbmann [27,28], Luca Settanni [29], Solveig Tosi [30,31], Benedetta Turchetti [8], Paola Visconti [32], Mirca Zotti [33] and Giovanna Cristina Varese [1,*]

1   Department of Life Sciences and System Biology, University of Turin, 10125 Turin, Italy;
    marino.moretti@unito.it (M.M.); jacopo.tartaglia@unito.it (J.T.); jolanda.perugini@unito.it (I.P.)
2   Institute for Sustainable Plant Protection, National Research Council (IPSP-CNR), 10135 Turin, Italy;
    gianpaolo.accotto@ipsp.cnr.it
3   Experimental Zooprophylactic Institute of Umbria and Marche "Togo Rosati" (IZSUM), 06126 Perugia, Italy;
    ms.beato@izsum.it
4   Department of Food and Drug, University of Parma, 43126 Parma, Italy; valentina.bernini@unipr.it
5   Department for Sustainability, Italian National Agency for New Technologies, Energy and Sustainable
    Economic Development (ENEA), 00196 Rome, Italy; annamaria.bevivino@enea.it
6   Experimental Zooprophylactic Institute of Lombardy and Emilia Romagna "Bruno Ubertini" (IZSLERBU),
    25124 Brescia, Italy; mariabeatrice.boniotti@izsler.it
7   Department of Agricultural Science, University of Sassari, 07100 Sassari, Italy; mbudroni@uniss.it
8   Department of Agricultural, Food and Environmental Sciences, Industrial Yeasts Collection DBVPG,
    University of Perugia, 06121 Perugia, Italy; pietro.buzzini@unipg.it (P.B.); benedetta.turchetti@unipg.it (B.T.)
9   INMI Biological Bank, Hospital National Institute of Infectious Diseases "Lazzaro Spallanzani" IRCCS,
    00149 Rome, Italy; stefania.carrara@inmi.it
10  National Institute of Oceanography and Applied Geophysics—OGS, 34010 Trieste, Italy; fcerino@ogs.it
11  Department of Medicine and Surgery, University of Milano-Bicocca, 20900 Monza, Italy;
    clementina.cocuzza@unimib.it
12  Agris Sardegna Agricultural Research Agency of Sardinia, 07100 Sassari, Italy; rcomunian@agrisricerca.it
13  Department of Medical Sciences and Public Health, University of Cagliari, 09042 Cagliari, Italy;
    scosenti@unica.it
14  Institute of Food Sciences, National Research Council (ISA-CNR), 83100 Avellino, Italy;
    antonio.dacierno@isa.cnr.it (A.d.); anna.reale@isa.cnr.it (A.R.)
15  Institute for Agri-Food Quality and Technology, Veneto Region's Agency for Innovation in the Primary Sector,
    36016 Thiene, Italy; paola.dedea@venetoagricoltura.org
16  Water Research Institute, National Research Council (IRSA-CNR), 28831 Verbania, Italy; laura.garzoli@cnr.it
17  Department of Life Sciences, University of Modena and Reggio Emilia, 42121 Reggio Emilia, Italy;
    maria.gullo@unimore.it
18  Department of Biotechnology, University of Verona, 37134 Verona, Italy; silvia.lampis@univr.it
19  Institute of Sciences of Food Production, National Research Council (ISPA-CNR), 70126 Bari, Italy;
    antonio.moretti@ispa.cnr.it (A.M.); giancarlo.perrone@ispa.cnr.it (G.P.)
20  Experimental Zooprophylactic Institute of Venezie (IZSVe), 35020 Padova, Italy; anatale@izsvenezie.it
21  Department of Environmental Biology, Sapienza University of Rome, 00185 Rome, Italy;
    annamaria.persiani@uniroma1.it
22  Experimental Zooprophylactic Institute of Piedmont, Liguria and Valle d'Aosta, 10154 Turin, Italy;
    monica.pitti@izsto.it
23  Institute of Biomolecular Chemistry, National Research Council (ICB-CNR), 80078 Pozzuoli, Italy;
    apoli@icb.cnr.it
24  Department of Biology, University of Naples "Federico II", 80126 Naples, Italy; antonino.pollio@unina.it
25  School of Agricultural, Forestry and Food Sciences, University of Basilicata, 85100 Potenza, Italy;
    annamaria.ricciardi@unibas.it

26   Institute of Agricultural Biology and Biotechnology, National Research Council (IBBA-CNR), 56124 Pisa, Italy; cristiana.sbrana@cnr.it

27   Department of Ecological and Biological Sciences, Tuscia University, 01100 Viterbo, Italy; selbmann@unitus.it

28   Italian National Antarctic Museum (MNA), Mycological Section, 16121 Genoa, Italy

29   Department of Agricultural, Food and Forest Sciences (SAAF), University of Palermo, 90128 Palermo, Italy; luca.settanni@unipa.it

30   Department of Earth and Environmental Sciences, University of Pavia, 27100 Pavia, Italy; solveig.tosi@unipv.it

31   National Biodiversity Future Centre, 90133 Palermo, Italy

32   Biological Resources Centre, IRCCS San Martino General Hospital, 16132 Genova, Italy; paola.visconti@hsanmartino.it

33   Department of Earth, Environmental and Life Sciences (DISTAV), University of Genova, 16132 Genova, Italy; mirca.zotti@unige.it

*   Correspondence: cristina.varese@unito.it; Tel.: +39-01-1670-5984

†   These authors contributed equally to the work.

**Abstract:** Microorganisms, microbiomes, and their products (e.g., enzymes, metabolites, antibiotics, etc.) are key players in the functioning of both natural and anthropized Earth ecosystems; they can be exploited for both research purposes and biotechnological applications, including fighting the big challenges of our era, such as climate change. Culture collections (CCs) and microbial Biological Resource Centres (mBRCs) are repositories of microorganisms that investigate and safeguard biodiversity and facilitate the scientific and industrial communities' access to microbial strains and related know-how by providing external users with skills and services. Considering this, CCs and mBRCs are pivotal institutions for the valorisation of microorganisms, the safeguarding of life, and the fostering of excellent bioscience. The aim of this review is to present the state-of-the-art of Italian CCs and mBRCs, highlighting strengths, weaknesses, threats, and opportunities. Italy is, indeed, a hotspot of microbial biodiversity with a high rate of endemism and incredible potential, not only for the food and beverage sector (i.e., "Made in Italy" products), where microorganisms can have a beneficial or a spoiling function, but also to guarantee environmental sustainability and foster the bioeconomy through the design of new bioprocesses and products. However, weaknesses, such as the lack of management rules in accordance with international quality standards, are also analysed and ways of overcoming them are discussed. In this context, an overview is given of the Joint Research Unit MIRRI-IT and the European-funded SUS-MIRRI.IT project, which aims to improve the management and sustainability of Italian microbial collections, and serves as a starting point for an innovative revolution in the context of CCs and mBRCs worldwide.

**Keywords:** microbial biobanks; biological resources; microbial community; microbial database; biodiversity; bioeconomy; microorganisms

## 1. Introduction

Microorganisms, including viruses, bacteria, archaea, yeasts, fungi, protozoa, and unicellular algae, are present in every natural and anthropized habitat, where they live and interact with their own ecosystems; they are key players in nutrient cycling through the decomposition and the mineralisation of organic matter, atmospheric nitrogen fixation, the solubilisation of the phosphate in the soil, and oxygen production. Moreover, they are also involved in some symbiotic relationships with plants and animals, where they play a pivotal role in their fitness and survival [1]. In addition, microorganisms are used in a wide range of human activities including food, medical, agricultural, veterinary, and biotechnological applications aimed at the production of both bulk and fine chemicals, e.g., biofuels, enzymes, vaccines, antimicrobials, antibiotics, and polysaccharides as well as fermented foods and beverages [2–9]. Consequently, due to their value, they are often referred to as "microbial resources". This term identifies all those microbial strains (including viruses) that are taxonomically defined, physiologically well characterised, genetically

stable, authenticated, well documented, quality controlled, and long-term preserved to allow their further application [10].

Nowadays, the increasing exploitation of natural resources and environmental disturbances due to climate change and anthropogenic activities, like intensive agriculture, livestock farming, and pollution, are accelerating ecosystem destruction with the consequent loss of microbial biodiversity [11,12]. The reduction of microbial resources is also a consequence of the extensive use of commercial starters in large-scale industrial and artisanal fermented foods and beverages [2]. The erosion of microbial biodiversity means not only the mere loss of species or biotypes but also the vanishing of potential biological tools that can be exploited to face problems such as the big challenges of the day [13]. In this framework, on one side, an increased knowledge of the ecological roles and functions of microorganisms and, on the other side, the safeguard of biological resources in microbial Culture Collections (CCs) can be considered strategic for developing new sustainable biotechnologies.

Microbial CCs are ex-situ repositories for biodiversity and the providers of microorganisms (living cells and their replicable parts), their genetic materials, and associated know-how. CCs that comply with a recognised Quality Management System (QMS) and follow the Best Practice Guidelines for the Conservation of Biological Resources published by the Organization for Economic Co-operation and Development (OECD), achieve microbial Biological Resource Centre (mBRC) status [14]. The main goals of CCs and mBRCs are the collection, characterisation, long-term preservation, and distribution to third parties of microbial strains, for both research and bio-industry purposes, along with the exchange of related metadata [15].

In the context of Open Science (i.e., making research results accessible to the whole scientific community), both CCs and mBRCs can be considered Open Science hubs due to their targets to supply and distribute microbial materials for scientific studies [16]. As such, CCs and mBRCs are, indeed, guarantors of legality since the microbial strains that are distributed must comply with international regulations (e.g., the Nagoya Protocol) and universally recognised biosafety and biosecurity protocols. Allowing access to this biological material means boosting future studies, facilitating new discoveries, and enabling verification and reproducibility of experiments and analyses. The increase in the citation of articles associated with strains deposited in public CCs highlights the positive impact of access to biological materials in terms of the accumulation of scientific skills [17]. Therefore, CCs and mBRCs will play an increasingly important role in the future as centres for biotechnological research, the conservation of microbial diversity, and the development of a sustainable bioeconomy.

The aim of this review is to depict the state-of-the-art of Italian CCs and mBRCs, highlighting their present strengths, weaknesses, threats, and future opportunities.

All presented information illustrates the scenario of "Italian available microbiota", which we hope will be of great use to any possible researcher in academia, industry, or interested stakeholder operating in the transition to a circular economy for the benefit of our planet.

## 2. An Overview of Italian Microbial Culture Collections

Currently, a number of Italian research institutions such as universities, CNR (National Research Council), national and regional agencies, hospitals, veterinary public health laboratories, and other public research institutions such as CREA (the Council for Agricultural Research and Economics) host microbial CCs. Overall, there are 31 main Italian CCs and 52% belong to universities, while 48% are hosted by public research institutions. Table 1 summarises the salient information of these CCs, which are distributed throughout the Italian territory (Figure 1). Noteworthy is that the majority of these CCs operate in more than a working field and some of them are extremely specialised and have a long-established tradition in a specific sector, i.e., the bio-restoration of cultural heritage, forensics, or veterinary research. Moreover, some CCs are particularly dedicated to the

preservation of a specific microbial category and possess historical outstanding knowledge and world-renowned expertise. This is the case, for example, for centres guaranteeing the preservation of microalgae, cyanobacteria, and plant and animal viruses.

**Table 1.** List of the major Italian microbial Culture Collections.

| Institution | Culture Collection | Location | Working Field(s) | Main Taxa Preserved | Ref. * |
|---|---|---|---|---|---|
| University of Basilicata | Unibas Yeasts and Bacteria Culture Collection (UBYC) | Potenza | Agriculture, Food | Bacteria, Yeasts | 1 |
| University of Cagliari | UNICA-DSMSP Collection (DSMSPC) | Cagliari | Agriculture, Environment, Nutrition | Bacteria, Filamentous fungi, Yeasts | 2 |
| University of Genoa | Culture collection of DISTAV-UNIGE (ColD) | Genoa | Environment, Nutrition, Pharma/Medicine, Bio-restoration of artistic heritage, Forensic | Bacteria, Filamentous fungi, Yeasts, Genomic DNA | 3 |
| University of Milano-Bicocca | UNIMIB Microbial Collection (MicroMiB) | Monza | Environment, Pharma/Medicine | Bacteria, Human viruses, Nucleic acids | 4 |
| University of Modena and Reggio Emilia | Unimore Microbial Culture Collection (UMCC) | Reggio Emilia | Agricultural, Food | Bacteria, Yeasts, Genomic DNA | 5 |
| University of Naples "Federico II" | Algal Collection at the University on Naples Federico II (ACUF) | Naples | Environment | Cyanobacteria, Microalgae | 6 |
| University of Palermo | UNIPA Microorganism Collection (UNIPAMC) | Palermo | Agriculture, Environment, Nutrition, Pharma/Medicine, Veterinary | Bacteria, Filamentous fungi, Yeasts, Cell lines | 7 |
| University of Parma | University of Parma Culture Collection (UPCC) | Parma | Agriculture, Food, Nutrition | Bacteria, Yeasts | 8 |
| University of Pavia | Amico Fungo | Pavia | Agriculture, Environment | Filamentous fungi, Yeasts, | 9 |
| University of Perugia | Industrial Yeasts Collection (DBVPG) | Perugia | Environment, Nutrition | Yeasts, Microalgae | 10 |
| Sapienza University of Rome | Fungal Biodiversity Laboratory (FBL) | Rome | Agriculture, Environment | Filamentous fungi | 11 |
| University of Sassari | Microbial collection of the University of Sassari (UNISSMC) | Sassari | Agriculture, Environment, Nutrition | Bacteria, Filamentous fungi, Yeasts | 12 |
| University of Turin | Turin University Culture Collection (TUCC) | Turin | Agriculture, Environment, Nutrition, Pharma/medicine, Veterinary | Bacteria, Filamentous fungi, Yeasts | 13 |
| University of Tuscia | Culture Collection of Fungi from Extreme Environments and National Antarctic Museum Culture Collection of Fungi from Extreme Environments (CCFEE) | Viterbo | Environment | Filamentous fungi, Yeasts | 14 |
| University of Verona | Verona University Culture Collection (VUCC) | Verona | Environment, Nutrition | Bacteria, Yeasts | 15 |

**Table 1.** *Cont.*

| Institution | Culture Collection | Location | Working Field(s) | Main Taxa Preserved | Ref. * |
|---|---|---|---|---|---|
| National Research Council (CNR) | ISA Culture Collection (ISACC) | Avellino | Food, Nutrition, Pharma/medicine | Bacteria, Yeasts | 16 |
| | IPSP Plant Viruses Italy (PLAVIT) | Turin/Bari | Agriculture, Environment | Plant viruses, Bacteriophages, Viroids, Plasmids, Polyclonal antisera, Infectious clones, Phytoplasmas | 17 |
| | Agro-Food Microbial Culture Collection (ITEM) | Bari | Agriculture, Environment, Food | Bacteria, Filamentous fungi, Yeasts | 18 |
| | Microbiology Lab IBBA Pisa Collection (MLIP) | Pisa | Agriculture, Environment, Nutrition | Bacteria, Filamentous Fungi, Yeasts, Microalgae | 19 |
| | Microalgae and extremophiles ICB Collection (ICBCC) | Pozzuoli | Chemistry, Environment | Microalgae, Archaea, Bacteria | 20 |
| | MEG-IRSA Culture Collection (MEGIC) | Verbania | Environmental | Bacteria, Filamentous fungi, Yeasts, Plasmids, Genomic DNA | 21 |
| National Agency for New Technologies, Energy and Economic Development sustainable (ENEA) | ENEA Culture Collection (ENEACC) | Rome | Pharma/Medicine, Nutrition, Agriculture, Environment, Bio-restoration of artistic heritage, Chemistry, Energy | Bacteria, Filamentous fungi, Yeasts, Microalgae, Plant viruses, Metagenomic DNA | 22 |
| National Institute of Oceanography and Applied Geophysics—OGS | Collection of Sea Microorganisms (CoSMi) | Trieste | Environment | Microalgae | 23 |
| Agricultural Research Agency of Sardinia (Agris Sardegna) | BNSS Culture Collection (BNSS) | Sassari | Agriculture, Environment, Nutrition | Bacteria | 24 |
| Veneto Agricoltura | Thiene Culture Collection (THCOLL) | Thiene | Agriculture, Environment, Nutrition | Bacteria, Filamentous fungi, Yeasts | 25 |
| IRCCS San Martino Polyclinic Hospital of Genoa | Interlab Cell Line Collection (ICLC) | Genoa | Pharma/Medicine | Cell lines | 26 |
| National Institute of Infectious Diseases (INMI) "Lazzaro Spallanzani" | INMI Collection (INMIC) | Rome | Pharma/Medicine | Bacteria, Sera, Nucleic acids, Plasma | 27 |
| Experimental Zooprophylactic Institute of Lombardy and Emilia Romagna | Biobank of Veterinary Resources (BVR) | Brescia | Nutrition, Veterinary | Bacteria, Animal Viruses, Cell lines, Sera, Histological material | 28 |
| Experimental Zooprophylactic Institute of the Venezie | IZSVe Veterinary Biobank (IZSVeB) | Legnaro | Nutrition, Veterinary | Bacteria, Animal Viruses, Filamentous fungi, Yeasts | 29 |

**Table 1.** *Cont.*

| Institution | Culture Collection | Location | Working Field(s) | Main Taxa Preserved | Ref. * |
|---|---|---|---|---|---|
| Experimental Zooprophylactic Institute of Umbria and Marche "Togo Rosati" | African Swine Fever Virus Collection (ASFVC) | Perugia | Nutrition, Veterinary | Bacteria, Filamentous fungi, Protozoa, Animal viruses, Sera, Genomic DNA | 30 |
| Piedmont, Liguria, and Valle d'Aosta Experimental Zooprophylactic Institute | Culture Collection of IZSPLV (IZSPLVCC) | Turin | Veterinary | Bacteria, Filamentous fungi | 31 |

* Collection or institute-hosting-collection Websites: 1 https://agraria.unibas.it/site/home/scuola.html (accessed on 16 February 2024). 2 http://www.mbds.it/portfolio/unica-dsmsp (accessed on 16 February 2024). 3 https://distav.unige.it (accessed on 16 February 2024). 4 https://www.unimib.it/ricerca/infrastrutture-ricerca/infrastrutture-europee-esfri/mirri (accessed on 16 February 2024). 5 https://www.umcc.unimore.it (accessed on 16 February 2024). 6 http://www.acuf.net (accessed on 16 February 2024). 7 https://www.unipa.it/dipartimenti/saaf (accessed on 16 February 2024). 8 https://www.foodproject.unipr.it/en/research/special-projects-at-unipr/the-university-microbial-collection-university-of-parma-culture-collection/64 (accessed on 16 February 2024). 9 https://terraeambiente.dip.unipv.it/it/dipartimento/risorse/laboratori-e-facilities/laboratorio-di-micologia-ambientale (accessed on 16 February 2024). 10 https://dsa3.unipg.it/DBVPG (accessed on 16 February 2024). 11 https://web.uniroma1.it/dip_dba302/en (accessed on 16 February 2024). 12 http://www.mbds.it/ (accessed on 16 February 2024). 13 http://www.tucc.unito.it (accessed on 16 February 2024). 14 https://steu.shinyapps.io/MNA-generale/ (accessed on 16 February 2024). 15 https://www.dbt.univr.it/?ent=iniziativa&id=9678 (accessed on 16 February 2024). 16 https://www.isa.cnr.it/web (accessed on 16 February 2024). 17 http://www.ipsp.cnr.it/plant-virus-italy-plavit-la-collezione-di-virus-vegetali-in-italia/ (accessed on 16 February 2024). 18 http://server.ispa.cnr.it/ITEM/Collection (accessed on 16 February 2024). 19 https://ibba.cnr.it/wp-content/uploads/2023/09/CNR-MLIP-EN-3.pdf (accessed on 16 February 2024). 20 https://www.icb.cnr.it (accessed on 16 February 2024). 21 http://www.meg.irsa.cnr.it (accessed on 16 February 2024). 22 https://www.enea.it (accessed on 16 February 2024). 23 https://cosmi.ogs.it (accessed on 16 February 2024). 24 https://www.sardegnaagricoltura.it/innovazionericerca/agris/ (accessed on 16 February 2024). 25 https://www.venetoagricoltura.org/2007/01/sedi/istituto-per-la-qualita-e-le-tecnologie-agroalimentari-thiene-vi/ (accessed on 16 February 2024). 26 http://www.iclc.it (accessed on 16 February 2024). 27 https://www.inmi.it/servizio/laboratorio_di_microbiologia_e_banca_biologica (accessed on 16 February 2024). 28 https://www.izsler.it/ (accessed on 16 February 2024). 29 https://biowarehouse.net/index.php/izsve/ (accessed on 16 February 2024). 30 https://www.izsum.it/ (accessed on 16 February 2024). 31 https://www.izsplv.it (accessed on 16 February 2024).

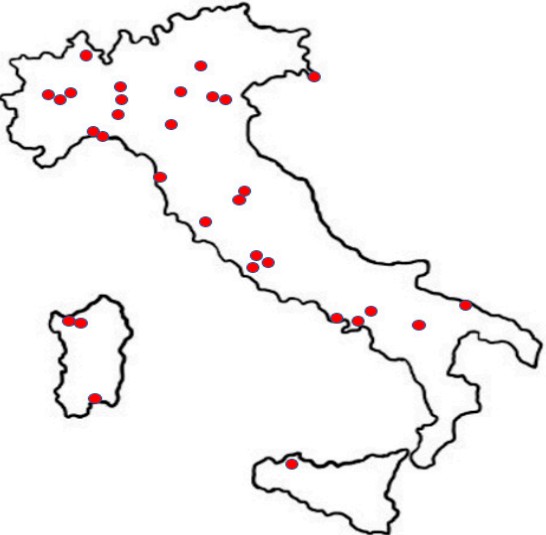

**Figure 1.** Geographic locations of major Italian microbial CCs evaluated in this review. Dots represent cities in which the 31 microbial CCs are operating. Note that multiple microbial CCs may be located in the same city.

This high number of CCs might be explained by the fact that Italy is a hotspot of microbial diversity, which is extensively exploited in food and beverage technologies—and

which characterise some "Made in Italy" products—as well as in medical, pharmaceutical, veterinary, and agricultural applications. Moreover, like other EU countries (e.g., Belgium), the Italian institutions hosting CCs joined a network that has guaranteed the coexistence of different microbial repositories dedicated to sundry types of microorganisms.

The different sectors of activity of Italian CCs are reported in Figure 2. Major working fields are related to, in order of importance, nutrition (such as starters for fermented food and beverages and probiotics), the environment (e.g., strains to be used in the bioremediation or biodegradation of complex organics compounds), agriculture (e.g., biological control agents, biofertilisers, and biostimulants), pharmaceuticals/medicine (e.g., preservation/production of isolates to be used for studying antibiotic production, resistance mechanisms, or virulence studies of clinical isolates), and veterinary (e.g., the identification/detection of animal pathogens and the production of functional feedstuffs).

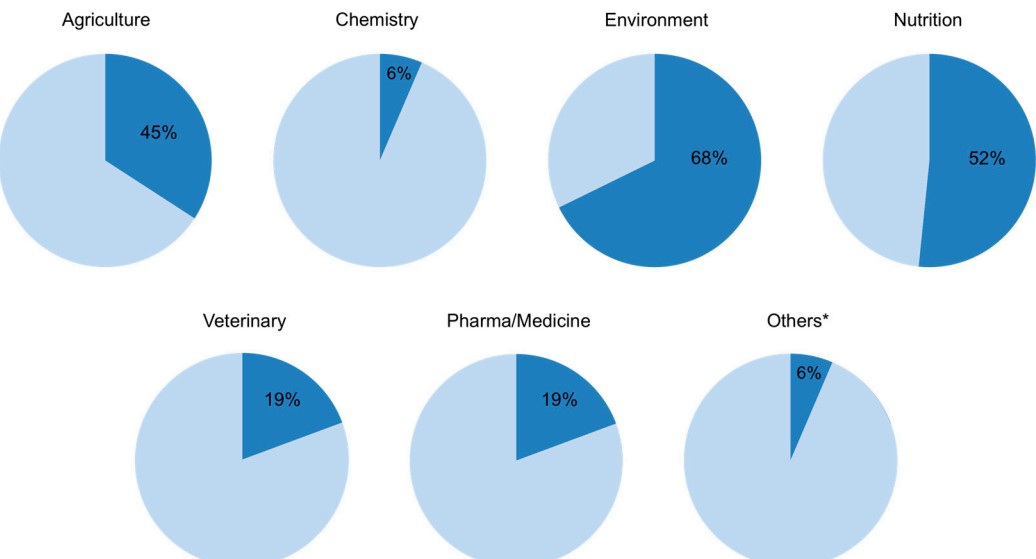

**Figure 2.** Working fields of major Italian microbial CCs. For each field, the percentage of CCs whose activities fall in that sector is shown. * Bio-restoration of cultural heritage, forensics, and energy production.

The heterogeneity of collected microorganisms is a key feature of the Italian network of CCs. Figure 3 shows the relationship between the type of preserved microorganisms and the number of collections that preserve them. The most represented microorganisms are bacteria, yeasts, and filamentous fungi. In contrast, archaea and cyanobacteria are preserved in only one CC. All other microbial groups are represented in at least two CCs. Some CCs preserve animal, human, and plant viruses as well as bacteriophages. Moreover, plasmids, sera, cell lines, nucleic acids, and other materials (indicated as "Others" in Figure 3) are also preserved.

Although CCs operate formally as independent institutions, often they are part of national or international networks, which promote collaboration and the exchange of ideas and information on all aspects of CC activities (from management to scientific aspects). Networking makes it possible to create synergies in addressing commonly faced challenges, e.g., the adoption of appropriate quality control systems, biosafety rules, and legal procedures, as well as the need for taxonomic, systematic, and bioinformatic expertise [18]. The vast majority (90%) of Italian microbial CCs are members of at least one network, such as the European Culture Collections Organisation (ECCO) and the World Federation for Culture Collections (WFCC) [19,20].

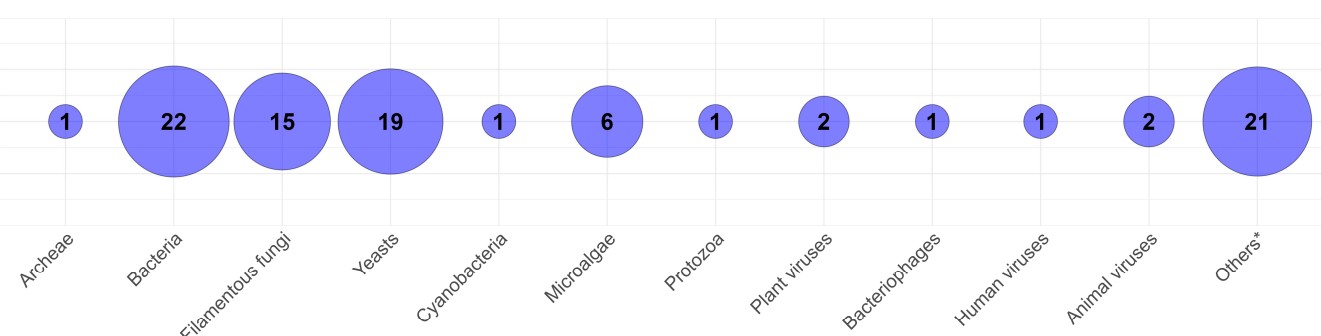

**Figure 3.** Bubble plot displaying the number of Italian CCs preserving microorganisms. Values in the bubbles represent the number of CCs. * Cell lines, plasmids, antibodies, antisera, nucleic acids, plasma, histological materials, infectious clones, and phytoplasma.

An EU networking initiative is the Microbial Resource Research Infrastructure–European Research Infrastructure Consortium (MIRRI–ERIC), which provides support to microbial CCs aiming to achieve mBRC status through the implementation of management standards and certification procedures [21]. The MIRRI–ERIC is, indeed, a coordinated network of EU mBRCs that deals with the conservation, characterisation, and distribution of microbial resources to the scientific community with an excellent standard of quality. Currently, the MIRRI–ERIC brings together more than fifty mBRCs from ten European countries. To date, Belgium, France, Latvia, Portugal, Greece, and Spain are founding members; Italy, the Netherlands, and Poland are prospective members; while Romania is an observer member. The MIRRI–ERIC possesses a hierarchical organisation with a Central Coordination Unit (CCU) located in Portugal that serves as the Executive Management Office, while every member is coordinated by a National Node that serves the needs of the local national microbial communities.

## 3. Joint Research Unit MIRRI–IT and the Project SUS-MIRRI.IT

A first attempt to organise the Italian microbial CCs network was made in 2017 with the establishment of a Joint Research Unit (JRU), named MIRRI–IT (Microbial Resource Research Infrastructure–Italy), among several universities and research institutes—namely, the Universities of Turin, Perugia, Modena, and Reggio Emilia, the San Martino University Hospital of Genoa, and the National Research Council (CNR) [22]. The mission of the JRU MIRRI–IT was mainly to overcome the discontinuity in the availability of resources and know-how provided by the Italian CCs, improving their quality management systems and focussing on the needs of stakeholders operating in the biotechnology industry. Over time, 22 new Associated Members have joined this network, hosting distinct new collections, and, currently, 31 microbial CCs are part of MIRRI–IT.

The activities of the JRU are regulated by an agreement subscribed to by all members. The JRU operates through the Coordination Group, the General Assembly of the Partners, and the Scientific Committee. MIRRI–IT guarantees the conservation and valorisation of an enormous number of biological resources (around 100,000 microbial strains).

The standardisation of resources and services supply provided by Italian microbial CCs is one of the main objectives pursued by MIRRI–IT together with the improvement of the quality management system, the implementation of biotechnological transfer, and the facilitation of entering the European network (MIRRI–ERIC) according to the dictates of the Partner Charter (a set of requirements to adopt and procedures to follow for participatory members). MIRRI–IT activities are organised into the following seven working groups: (i) microbial identification, characterisation, and application; (ii) microbiomes; (iii) common procedures and protocols; (iv) collection data management; (v) biosecurity; (vi) national plan on biodiversity of agri-food interest; and (vii) promotion and communication.

The JRU is, in addition, responsible for managing the relationships and institutional interactions of the microbial CCs with government bodies at regional, national, and European

levels as well as with other research centres. The promotion activity in both the academic and industrial fields and, more generally, communication with civil society, represents other key elements of the JRU's strategic plan. Finally, noteworthy is that the JRU coordinates the participation of Italian microbial CCs in various international projects and funding calls, including the recently awarded EU project "Strengthening the MIRRI Italian Research Infrastructure for Sustainable Bioscience and Bioeconomy" (SUS-MIRRI.IT).

SUS-MIRRI.IT is a project funded by the "NextGenerationEU" programme through the Italian "National Recovery and Resilience Plan (PNRR) Italiadomani" [23]. The project was launched in 2022, with a total budget of about 17 million euros, with the primary aims of implementing the supply of microbial resources conserved in Italian CCs, improving their characterisation, optimising their management, and, therefore, making the most of their genomic and metabolic potential for biotechnological and industrial purposes. It includes 20 microbial CCs distributed all over the Italian territory, the majority of which belong to the JRU MIRRI–IT. The project is coordinated by the University of Turin and involves 15 national institutions divided into 24 Operational Units (OUs). SUS-MIRRI.IT is organised into six Work Packages (WPs) (Figure 4); the implementation and harmonisation of the quality management of Italian microbial CCs will be mainly achieved by empowering CCs in terms of new laboratories equipped with cutting-edge technologies for the conservation and characterisation of microbial resources. Furthermore, the definition of several common procedures, inspired by the biobank ISO standard 20387 [24], which should be adopted by all microbial CCs to improve their quality standard, has been set. These procedures related to the acquisition of microorganisms and associated data, quality control, storage, and distribution of microorganisms should enable Italian microbial CCs to reach the mBRC level required to join the MIRRI–ERIC and present Italy as an international player in the responsible management of biological resources. With regards to data management, the project is dedicated to the development of an online platform containing the integrated catalogue of all Italian microbial resources (ItCCC), which aims to overcome the lack of a unique access point to the microorganisms, associated metadata, and services. This will valorise the stored resources, increasing their visibility and optimising their access. Moreover, the creation of dedicated software will support the rational governance of each microbial CC. Open-source workflows have also been created for bacteria, fungi, and virus genome assembly, annotation, and whole-microbiome description. The project will also address challenges in the use of microbiomes as new frontier tools to face current and future critical issues in various fields such as agriculture, environmental protection, food production, the bio-restoration of artistic heritage, pharmaceuticals, green chemistry, and animal, plant, and human health. In detail, it will try to fill the gap regarding the modalities through which microbiomes are sampled, characterised, securely stored, and subsequently used for further applications, producing new knowledge [25]. Finally, the main efforts are devoted to diffusing achievements to the microbiological community and to instil, even in civil society, a consciousness about the enormous potential of microbial resource exploitation.

In the end, besides promoting scientific advancement, technological innovation, and the competitiveness of Italian microbial CCs, the project SUS-MIRRI.IT expects to bring beneficial impacts at a higher scale across health security (via supporting the design of new technologies for the prevention, early diagnosis, and treatment of pathogens including the production of vaccines, monoclonal antibodies, highly sensitive diagnostic kits, phage therapies, and gut microbiome transplants) to food production (via inspiring industrial research into functional foods, probiotics, and nutraceuticals) and environmental protection (for example, by developing new eco-friendly and sustainable solutions for agriculture and soil management or the production of new materials useful for bioenergy production).

Finally, it should be mentioned that all data presented in this manuscript have been collected and collated via two surveys prepared and distributed to the major Italian CCs within the scope of the JRU MIRRI–IT and the SUS-MIRRI.IT project. The questionnaires focussed on general information about the microbial CCs, the Quality Management System,

database organisation, and offered services and training courses. The analysis of the questionnaires made possible the identification of the strengths of the microbial CCs as well as the weaknesses and potential threats they face, to which attention will be paid in the future to improve and standardise to meet international requirements.

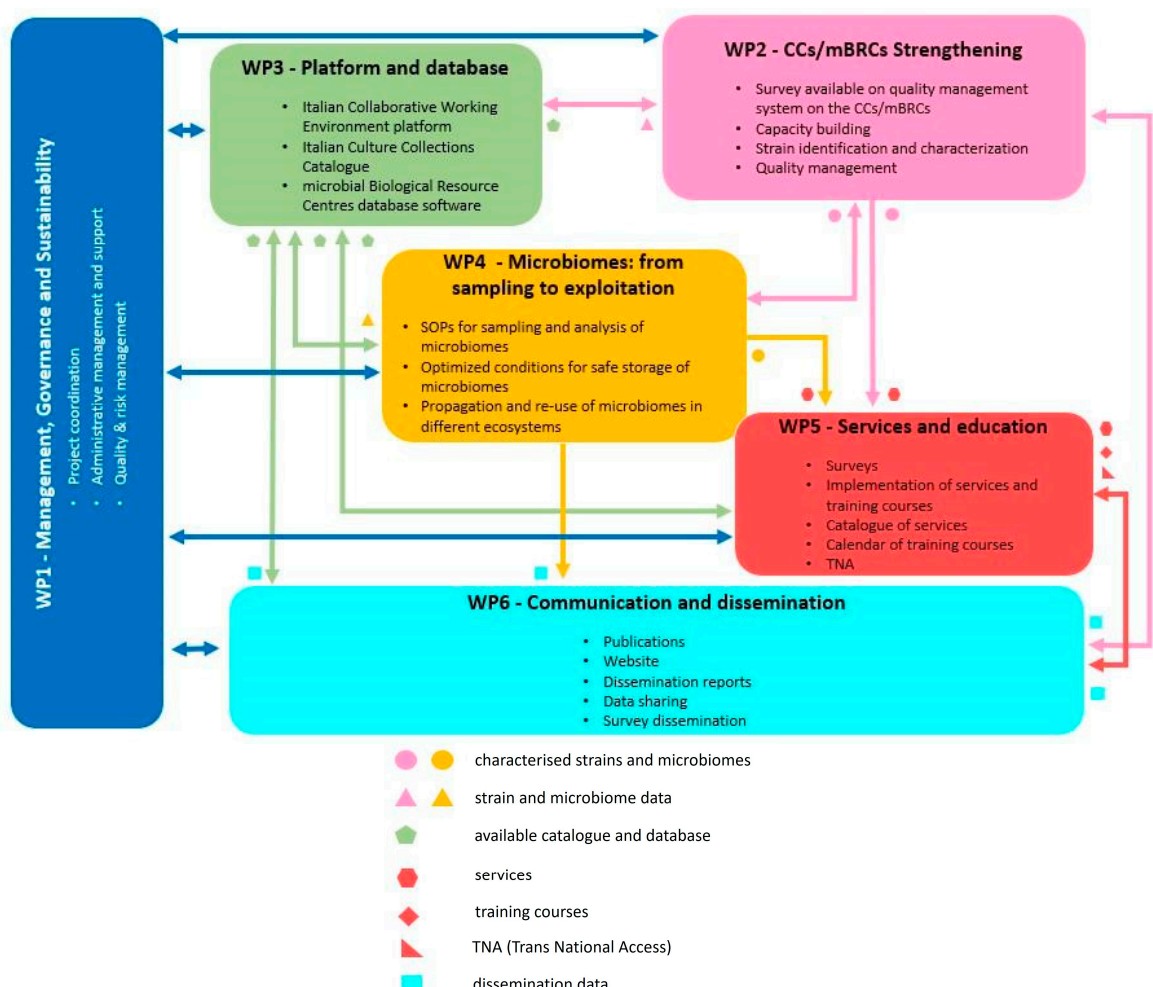

**Figure 4.** PERT chart of the SUS-MIRRI.IT project. Interrelations among WPs are depicted using two-way or one-way arrows according to bidirectional or unidirectional impact, respectively. Arrow colour is determined by the WP of reference, giving priority to those with lower numbering in the cases of bidirectional arrows. WP products are represented by different adjacent symbols, having the same colour as the reference WPs.

## 4. Microbial Resources and Service Supply

As already stated, the main goals of microbial CCs are the collection, characterisation, conservation, and distribution of microbial resources and related information.

Figure 5 displays the number of strains preserved in major Italian CCs organised according to macro-groups of microorganisms. To date, the total number of microbial strains amounts to 102,405 (excluding sera and tissue samples), which corresponds to over 7,000 microbial species from different kingdoms. Many microorganisms solely annotated with the "genus name" or higher taxonomic classification (meaning that the species name is not (yet) defined) are also present (Figure 6). Microorganisms included in this category might be of valuable interest for future characterisation. Notably, the numbers reported in Figures 5 and 6 only refer to strains available to customers, so they do not include microorganisms stored for particular purposes, such as patent or safe depositions, or all isolates preserved in minor CCs and research institutes, which are not considered here.

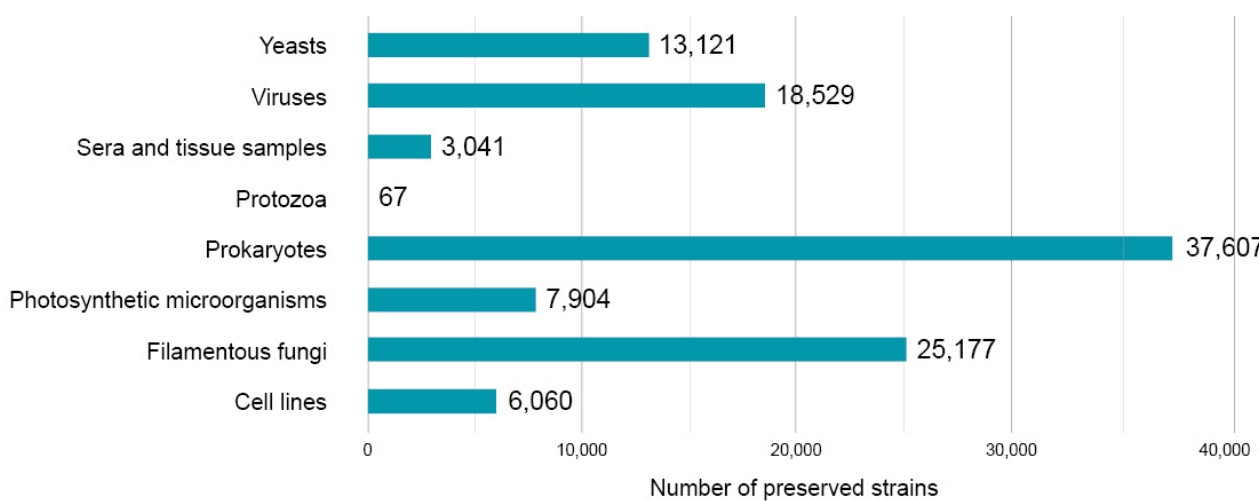

**Figure 5.** Number of microbial strains publicly available in major Italian CCs. Cyanobacteria and microalgae are grouped together as photosynthetic microorganisms. Viruses of different hosts (bacteria, plants, animals, and humans) are considered under the same group. Prokaryotes include bacteria and archaea.

**Figure 6.** Characterisation of microorganisms stored in major Italian microbial CCs. Microbial resources are categorised according to species identification. Names of the collections refer to those defined in Table 1.

Although, as expected, an overlap in the species preserved in the evaluated Italian microbial CCs is present, the total number of strains is considerable when compared to

the ca. 31,200 microbial entries preserved in the German Collection of Microorganisms and Cell Cultures (DSMZ), the ca. 76,000 entries in the American Type Culture Collection (ATCC), and the ca. 130,000 entries declared by the Westerdijk Fungal Biodiversity Institute (WI-KNAW, formerly CBS) [26–28].

Different types of deposits exist in CCs or mBRCs. A "Public Deposit" means that the deposited biological material remains available to the public. These microbial resources can be used for scientific research or the description of new species. According to the International Code of Nomenclature of algae, fungi, and plants, when a new species is described, it is strongly recommended to deposit the type-strain in at least two different and physically separate institutional CCs [29]. In comparison, for bacteria and archaea, following the International Code of Nomenclature of Prokaryotes, the description of new species requires the deposition of a living type-strain in two CC members of the WFCC, located in different countries [30]. About two-thirds (65%) of the Italian CCs preserve type-strains or strain descendants of the original isolates used in species description that exhibit all relevant phenotypic and genetic properties of the original taxonomic characterisation; to date, this includes 821 for bacteria, 636 for yeasts, 228 for animal viruses, 239 for filamentous fungi, 51 for plant viruses, 16 for microalgae, 4 for archaea, and 4 for cyanobacteria.

Microbial resources to which access is permitted only with prior authorisation from the depositor are stored as "Safe Deposits". Additionally, if CCs and mBRCs are recognised as International Depository Authorities (IDAs) and the storage of microorganisms is intended for the legal protection of intellectual property, the deposit is called a "Patent Deposit". IDAs must comply with a series of rules and regulations according to the requirements of the International Treaty signed in Budapest (Hungary) in 1977 [31,32]. Among the 31 main Italian microbial CCs analysed in the present review, 74% offer a Public Deposit of microorganisms, 34% a Safe Deposit, and 10% a Patent Deposit.

The identification and characterisation of biological resources are strictly necessary to guarantee the quality and the authenticity of preserved material [15]. Several approaches are used to correctly identify microorganisms, including molecular, morphological, serological, and biochemical characterisation. However, for a robust identification outcome, it is required to combine the previously mentioned methodologies, e.g., integrating the analysis of gene sequences with biochemical profiling (i.e., metabolites and/or extrolites) [15].

Morphological identification approaches are still often applied to organisms such as filamentous fungi, protozoa, viruses, and microalgae, which usually show a high shape diversity of anatomical traits (especially sexual structures) useful to discriminate them. However, nowadays, the analysis of morphological traits is associated with at least the molecular approach to reach the species identification level (especially for filamentous fungi).

Biochemical identification is mostly used in the clinical diagnostics of bacteria and yeasts. Various phenotypic screening tools based on biochemical properties, such as API ID32C and AuxaColor, are commercially available [33–36] as well as automated metabolic characterisation facilities, i.e., VITEK 2 and Biolog's Omnilog System [37,38]. Recently, MALDI-TOF mass spectrometry (based on the comparison of protein mass spectra with well-characterised reference strains included in reliable databases) is proving to be a commonly used methodology for microbial (mainly bacterial) identification, especially in the clinical fields but also in commercial sectors [39–41].

Molecular marker sequencing is the most widely used methodology to infer the identity of microbial strains. For most bacteria species (except Enterococcaceae), sequencing of 16S rDNA, either partial or total, is considered the "gold" standard procedure for taxonomic definition [42,43], while for fungi, the ITS region—although often this needs to be associated with additional genes (e.g., beta-tubulin, calmodulin, actin, *tef*, *rpb2*, etc.)—is required to obtain a more robust taxonomic definition [44–46]. The discrimination of closely-related bacterial species within genera is also achieved by gene sequencing, such as *rpoA* (α subunit of RNA polymerase), *rpoB* (β subunit of RNA polymerase), *recA* (recombinase A), *sodA* (manganese-dependent superoxide dismutase), *groEL* (60 kDa heat-shock protein),

*gyrA* (A subunit of DNA gyrase), *gyrB* (B subunit of DNA gyrase), and *vapA* (virulence associated protein A) [47,48].

For yeasts, the combined adoption of the D1/D2 domains of the LSU rDNA gene, together with the ITS region, is considered an effective approach for the rapid identification of species and the isolation of evolutionary new lines [49].

For viruses and microalgae, molecular identification is mainly carried out on the genes of the coat protein and rbcL (the large subunit of the $CO_2$-fixing enzyme RuBisCO) or 18S and 28S ribosomal genes, respectively [50,51].

Genome-wide genotyping techniques, also known as typing methods, are also used for the identification and characterisation of microbial resources. Some of these methodologies, although developed more than 30 years ago, are still in use, e.g., Random Amplification of Polymorphic DNA (RAPD), Repetitive extragenic palindromic PCR (Rep-PCR), Restriction Fragments Length Polymorphism (RFLP), Amplified Ribosomal DNA Restriction Analysis (ARDRA), Pulsed Field Gel Electrophoresis (PFGE), and MultiLocus Sequence Typing (MLST) [42,52]. Most of these techniques can be efficiently used for the massive typing of isolates and the evaluation of the redundancy of strains, representing robust tools to support the work of CCs and mBRCs.

Today, Whole-Genome Sequencing (WGS) obtained by Next-Generation Sequencing (NGS) technology is mainly used to characterise the genomes of specific types of microorganisms or to perform "whole-genome comparison" analyses [42]. This approach provides a deeper level of phylogenetic and taxonomic knowledge as well as useful information on the metabolic, ecological, and, thus, biotechnological potential of microorganisms. Moreover, the evolution of this technology from short-read sequencing to long-read sequencing allows the detection of complex structural variants, such as large inversions, deletions, or translocations, that may be difficult to detect with short reads that mainly capture genetic variations only. Some of these variants have been implicated in areas like epigenetics, resistance to antibiotics, and virulence [53].

In major Italian CCs evaluated in this review, morphological and/or physiological techniques have been used, especially in the past, to identify—at least at the genus level—more than half (59%) of the microbial resources preserved. In detail, this encompasses 75% of filamentous fungi, 54% of bacteria, 22% of yeasts, 95% of viruses, 9% of microalgae, 49% of cyanobacteria, and 100% of protozoa and archaea.

As stated above, identification through molecular markers is the best discriminative method for microorganisms since it does not require complex or excessively expensive instrumentation. Nevertheless, a quite high number of strains preserved in Italian microbial CCs are still identified via techniques that do not use them. This is, for example, the case of viruses for which serological techniques are considered the fastest, most specific, cheapest, and most reliable methods.

Currently, characterisation via molecular techniques is offered by 43% of the major Italian microbial CCs, NGS by 24%, genotyping by 48%, virus detection and identification by 24%, and screening, testing, and biological assays by 52%.

Re-identification and additional characterisation of many preserved microbial strains through molecular methods are always ongoing, especially for those microorganisms whose previous identification was done when molecular techniques were not so common. Currently, 40% of filamentous fungi, 39% of bacteria, 47% of yeasts, 34% of viruses, 4% of microalgae, 25% of cyanobacteria, and 100% of archaea have been identified with molecular methods.

The fact that services concerning genotyping, genome sequencing, and molecular marker sequencing are offered by only half of the Italian CCs does not reflect the lack of necessary equipment or resources (human or financial) but, rather, the fact that the speed of evolution of the techniques makes the outsourcing of the service more convenient and economically advantageous.

Concerning biochemical characterisation, recently (and also due to the funding of the SUS-MIRRI.IT project), several CCs have acquired the necessary equipment for MALDI

analysis. Therefore, the identification of microbial strains (to date, only 2.5% of clinic bacteria stored in major Italian CCs have been identified using MALDI-TOF MS) via this rapid, accurate, and precise innovative technology is expected to exponentially increase in the coming years.

For the preservation of microbial resources, methods used by CCs and mBRCs can be divided into three types depending on the duration of the culture storage—short-term storage (S), which ranges from a few weeks to a few months, medium-term storage (M), which ranges from a few months to a few years, and long-term storage (L), which encompass several years. Without doubt, the longer the cultures remain in a quiescent state, the lower the cell viability; therefore, periodic quality checks are necessary to verify the preservation conditions of the material. The most common ex-situ preservation techniques are subculture on solid media (S), agar beads, or gelatine discs (M); storage in sterile soil (M); storage in sterile water (M); storage in mineral oil (M); storage in silica gel (M); spray-drying (M); liquid-drying (M); desiccation (M); cryopreservation at −20/−80 °C or −140/−152 °C (L); freeze-drying (L); and vitrification (L) [54,55]. Regarding the preservation of biological material over long periods (i.e., years), cryopreservation and freeze-drying, (lyophilisation) are the most commonly used methods since the metabolism of microorganisms is extremely reduced if not completely stopped [56,57]. Under these conditions, microbial cultures remain in a state of quiescence that can last for several years. In addition, these techniques do not require the use of fresh culture media and regular subcultures to keep the material alive [56]. Moreover, cryopreservation and, in particular, freeze-drying allow biological material to be stored in a small space, avoiding contamination and maintaining genetic stability [56]. However, the effectiveness of such techniques is limited for certain microorganisms, or their "replicable parts", as some bacteria, cyanobacteria, fungi (especially non-sporulating fungi), and microalgae may suffer cellular damage during freezing or drying, leading to poor viability or death [58,59]. Consequently, for many microbial species, it is necessary to use and develop ad-hoc cryopreservation and lyophilisation protocols based on empirical evidence and quality-control measures to ensure reproducibility and reliability [60,61]. The main conservation techniques for microbial resources used in major Italian CCs are cryopreservation at −20/−80 °C, subculture on a solid substrate, and freeze-drying. Additional methods are reported in Figure 7.

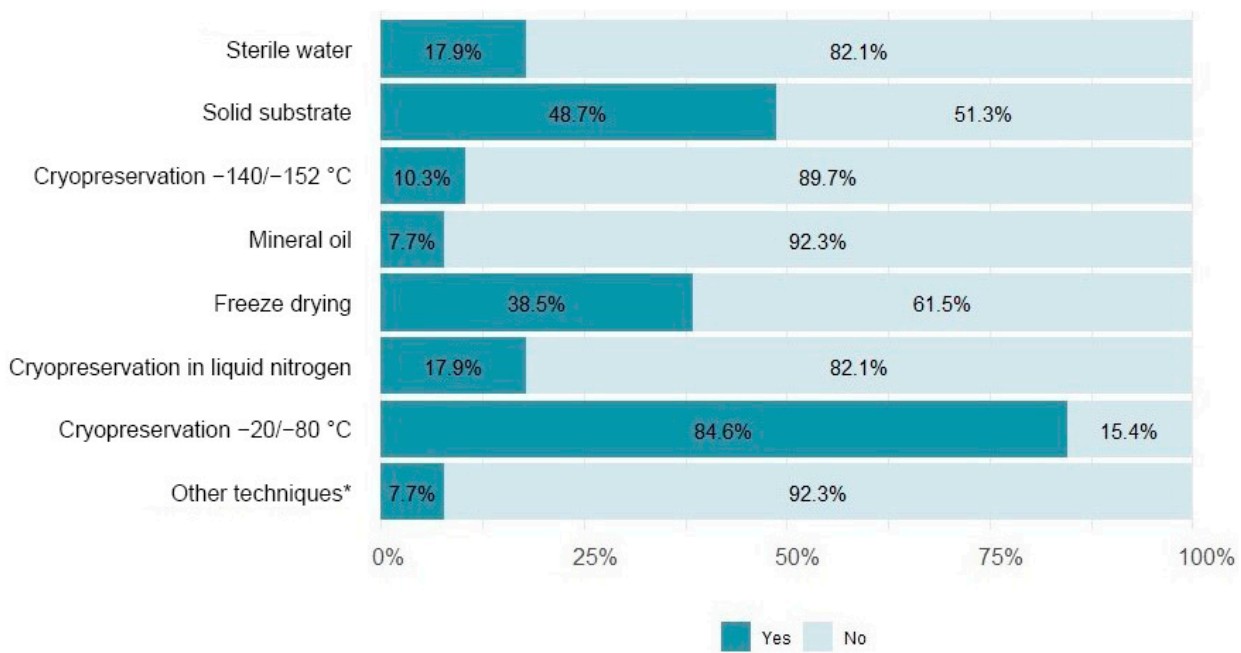

**Figure 7.** Preservation techniques used by major Italian CCs to store microbial resources. Percentages of usage of each method are displayed. * Solid plugs in glycerol, vitrification, silica gel, and on paper discs.

In agreement with WFCC and OECD guidelines, it is highly recommended to use at least two different methods for storing microbial cultures. The use of different storage strategies ensures the long-term persistence and stability of the culture, to minimise the risk of accidental loss or potential death [14,62]. Only four major Italian microbial CCs out of thirty-one use at least two preservation techniques for all stored strains. In contrast, the majority use only one preservation technique on all or up to 75% of the stored strains. Concerning additional storage in a separate location, the data are extremely worrying; only 26% of CCs possess a second safe storage facility physically distant from the principal hub. This serious deficiency means that Italian microbial CCs, although rich in preserved biological material, are extremely prone to accidental loss, which represents a severe weakness to address. As well as the implementation of preservation techniques, a main objective of the SUS-MIRRI.IT project, as cited above, is, indeed, the set-up of a disaster plan for CCs or a protocol describing the strategies, actions, and technological tools necessary to restore the correct functioning of a collection after the occurrence of an emergency or unexpected negative event. Among these actions, backup storage is one of the main foreseen intents undertaken by the Italian microbial CCs.

Quality control of the stored material is a key procedure to assess and ensure the conservation status of the microorganisms in the collection. Quality controls include the assessment of viability, purity, identity, and phenotypic stability of the stored material [56]. The main controls performed by the Italian microbial CCs are viability, purity, and re-authentication assessments of microbial strains using morpho-physiological or molecular techniques. Checks are generally performed at four different points, which are (i) upon receipt of the biological material, (ii) immediately after storage, (iii) during long-term storage, and (iv) before distribution. Test experiments are also carried out to check for the absence of contaminants. Figure 8 summarises the situation for Italian microbial CCs with respect to carrying out quality-control checks according to the stages of the preservation of biological material.

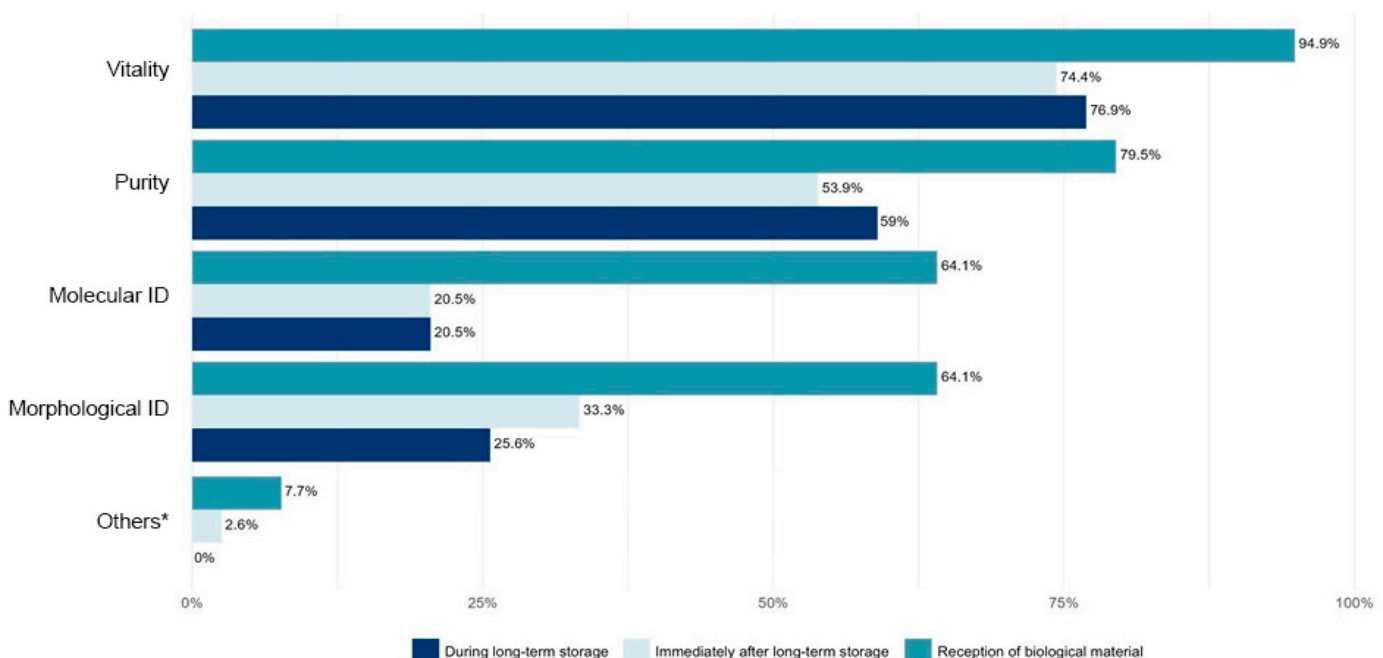

**Figure 8.** Quality control checks performed by major Italian microbial CCs on preserved material at different time points, from the acquisition of the microorganism to the long-term storage. The percentages shown are related to the collections offering the service of Public Deposit. * Serological ID, concentration/quantitative check, etc.

As well as being centres for the preservation of microbial resources, CCs also constitute research infrastructure providing services and expertise to third parties. External users are not only (national and foreign) research institutions but also companies, mainly small/medium enterprises [22]. In this way, CCs and mBRCs can play the role of promoters of scientific progress and also stimulate the development of the bio-industry in the country or geographical region in which they operate. Moreover, centres that are able to provide services for a fee can have an additional financial income. In this fashion, they do not only rely on internal financial resources, often derived from public funding. Generally, the main services offered by CCs and mBRCs are the storage of microbial strains, the distribution of biological and/or genetic material together with the associated information and metadata, the identification and characterisation of microorganisms, and the isolation of microorganisms from various matrices or samples. In addition, CCs and mBRCs can deliver training courses on different microbiology-related topics. As the demand for Education and Training (E&T) services has increased in the last few years [63], CCs and mBRCs could play a central role in implementing the training of new researchers in academia or companies or personnel working for public authorities involved in science policymaking. Therefore, they are acquiring a true leading position in spreading scientific knowledge and academic know-how.

Advanced training courses provided both face-to-face and virtually (as webinars) by Italian microbial CCs are displayed in Figure 9.

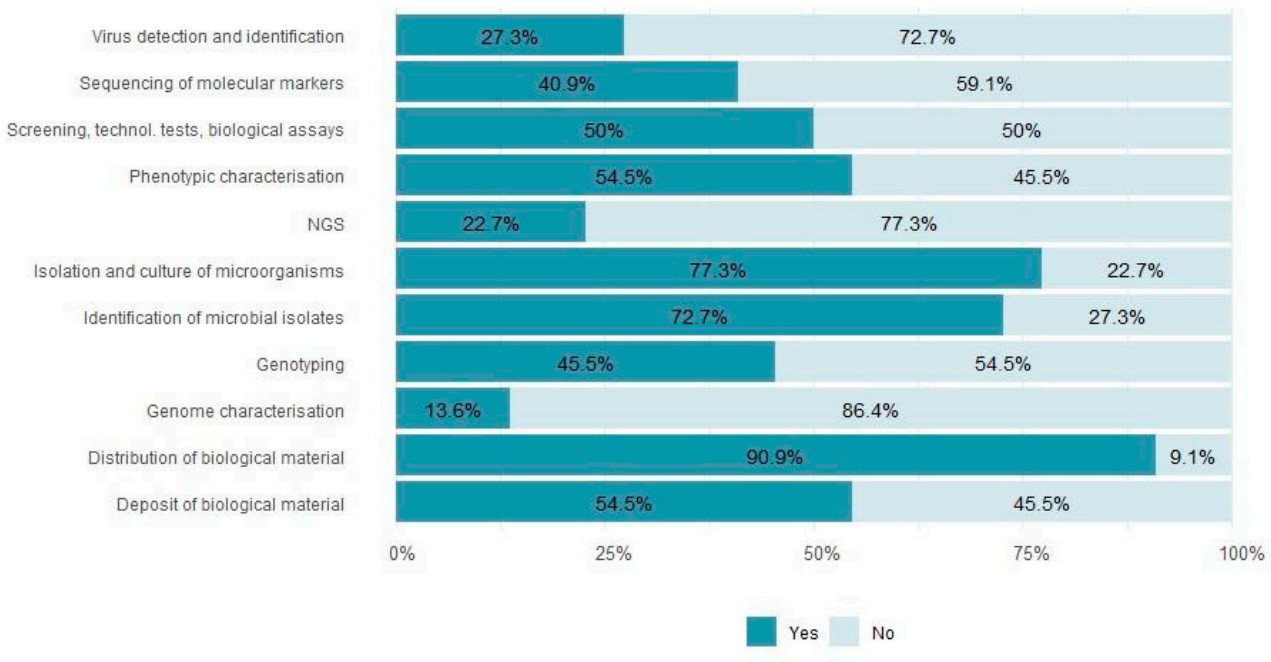

**Figure 9.** Advanced training courses offered by major Italian CCs. The percentages shown are related to the collections offering the services.

In Italy, 68% of the microbial CCs offer external services and 90% provide these services for a fee. It is not possible to define an average price for the various services as much depends on the scientific instrumentation on which the centres rely and on the type of microorganism for which the service is requested. For example, there are strains whose distribution costs more than others for several reasons, such as difficulty of preservation or transport, abundance in the collection, high rate/probability of contamination, etc.

Figure 10 shows the main types of services and the percentages of microbial CCs providing them; the figure depicts the situation for the last five years, without considering the COVID-19 pandemic period, which strongly affected CC activities.

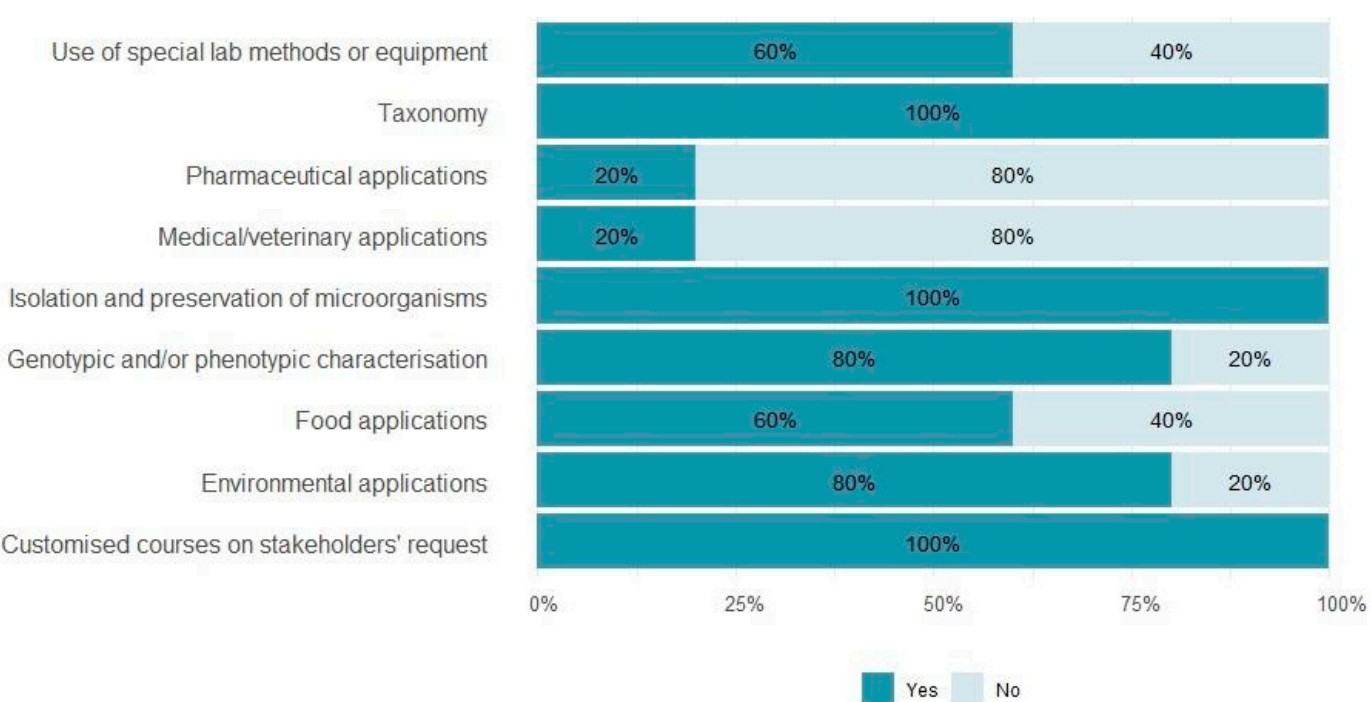

**Figure 10.** Service supply of major Italian microbial CCs. Main provided services are listed. The percentages shown are related to the collections offering the service.

Concerning isolation and culture services, microorganisms can be isolated/purified from mixed cultures or from different matrices such as soil, water, food, biological materials, etc, and storage conditions can be optimised alongside microbial cultivation. It is worth noting that the fermentation services delivered by several Italian microbial CCs are being scaled-up. Due to equipment such as bioreactors, fermenters, and bioprocess systems, users—especially industries—can customise their need for microbial biomass, enzymes, proteins, polysaccharides, or other biomaterials to be used in industrial reactions.

Moreover, an estimation of the number of cells/viruses present in a culture or sample can be performed.

Finally, groundbreaking work is being performed by several Italian CCs on sampling, characterisation, and preservation of microbiomes. The study of microbiomes is still in its infancy, but the whole microbiological community worldwide knows that unravelling complex interactions between microorganisms in microbiomes will profoundly impact many aspects of human health, disease prevention, agricultural productivity, food production, environmental sustainability, and even forensic science, thereby opening new frontiers in science [64,65].

## 5. Management of Italian Microbial Culture Collections

Among the most important challenges that microbial CCs have to face nowadays are (i) compliance with national and international legal frameworks (e.g., the Convention on Biological Diversity, the Nagoya Protocol on access and benefit-sharing, quarantine lists, packaging and shipping regulations, etc.), (ii) the implementation of the Quality Management System (QMS) that should always keep internal operations updated, (iii) ensuring digital information and technology access guided by the FAIR (findable, accessible, interoperable, and reusable) principles, (iv) the pursuit of a continuous amended capacity building and the training of expert personnel, (v) the application of new technologies to maintain state-of-the-art know-how and always meet the expectations of both customers and stakeholders, and (vi) future financial sustainability.

The effective management of microbial CCs is fundamental for the supply of authenticated microbiological materials, associated metadata, and services with guaranteed,

harmonised international high-quality standards. Considering this, many CCs have introduced certification/accreditation for quality to achieve preferred supplier status, maintain the search for excellence, and increase competitiveness.

Management guidelines for microbial CCs have been developed by many organisations, e.g., the WFCC, ECCO, OECD, ISO (International Standards Organization), WHO (World Health Organization), and national authorities such as the French Euro-Quality System (Standard NF S96-900) [14,62,66–69]. International projects, from which European research infrastructures such as the EMbaRC (European Consortium of Microbial Resources Centres), BBMRI (Biobanking and BioMolecular Resources Research Infrastructure), EVA-GLOBAL (European Virus Archive GLOBAL), and IS-MIRRI21 (Implementation and Sustainability of Microbial Resource Research Infrastructure for the 21st Century) emerged, have also provided support for the development of standards and rules applicable to microbial CCs [70–73].

Compliance with these guidelines and rules implies the adoption of a rigorous code of conduct based on a continuous quality evaluation over time assessed through process review, the analysis of costs and performance, the allocation of resources, personnel competence, consistency in the services offered, the satisfaction levels of stakeholders, external audits, benchmarking, and opportunities for improvement.

In addition to internal management rules and quality policy, CCs should also comply with legislation regulating biosafety and biosecurity to protect public health and the environment from accidental exposure to dangerous microorganisms and to prevent misuse [74]. This includes intellectual property rights (such as the Budapest Treaty regulating the process of patent deposits); traceability and legality of the microbial resources acquired; distribution (such as the Nagoya Protocol and EU Regulation N. 511/2014 aimed at the fair and equitable sharing of benefits derived from the use of genetic resources); and norms specifically addressing genetically modified organisms (GMOs) [32,75–77].

The management situation of microbial CCs in Italy is very varied and reflects the extreme territorial fragmentation. A great attempt to achieve standardisation—based on certified internationally recognised quality criteria—has been made in the last 5 years by the JRU MIRRI–IT and is currently one of the main aims of the SUS-MIRRI.IT project.

The analysis of the surveys distributed to the major collections shows that almost half of them (48%) follow international management guidelines but only a third (29%) work according to a certification/accreditation system (mainly ISO standards).

Notably, almost half of the CCs (45%) use a Material Accession Agreement (MAA) or Material Deposit Agreement (MDA) regulating the deposit of biological material into the collection, while only one quarter (26%) takes account of the Nagoya Protocol (transparency on procedures for access and benefit-sharing with the country of origin of the biological resource) when incorporating biological materials into their collection. Concerning distribution, most CCs (74%) use a Material Transfer Agreement (MTA) to provide microorganisms to users. Special care and permissions are then devoted to quarantine, dual use (illicit use for harmful purposes), and GMO stored microorganisms.

In terms of biosecurity and biosafety, half of the major Italian CCs (54%) are classified as BSL2 (preserving pathogenic or infectious organisms that pose a moderate health hazard, e.g., *Staphylococcus aureus* causing staphylococcal infections); 34% are BSL1 (preserving low-risk microbes that pose little to no threat of infection in healthy adults, e.g., non-pathogenic strains of *Escherichia coli*), and 12% are BSL3 (preserving microbes, either indigenous or exotic, that can cause serious or potentially lethal disease through inhalation, e.g., bacteria causing tuberculosis). According to their category, CCs adopt biorisk management guidelines, such as limited and controlled access to the physical storage place of the material, the traceability of the biological resources in all procedures, and specific training of the personnel working in the collection, becoming gradually more restrictive with each category. The latter aspect is crucial for the proper steering and sustainability of CCs as human capital requires continuous updating with the always newly-released technologies,

rules, information, and applications in the field of microbial deposition, patenting, rights protection, and related areas.

The proper training of personnel to provide them with the skills and knowledge needed to perform the essential tasks necessary for the everyday running of the collections is a high priority for the performance of a microbial biobank. Unfortunately, there is a high turnover of staff in CCs as people are rarely employed full-time or permanently, with the consequence that investment made in training and mentoring is a loss rather than ensuring a gain or return on investment. From an accurate evaluation of the major Italian CCs, 70% resulted, indeed, employing <6 person/months (a metric for the time (effort) that the personnel working in the collection devotes to its management, where a 1 person/month corresponds to the actual annual productive hours divided by 12), while 20% had less than 12 person/months and only 10% can guarantee >12 person/months.

Considering this, it should be no surprise that the majority of Italian microbial CCs do not possess adequate IT expertise, including the competence to run an online catalogue for microbial resources and services or a machine-readable catalogue where data are stored in XML, JSON, RDF, or similar file formats, which could be easily exported/manipulated with digital applications. From our analysis, only 15 collections out of 31 display an online catalogue; however, a similar trend is observed worldwide where, currently, only 53% of microbial CCs have a digital catalogue of preserved microorganisms (data retrieved from the World Data Centre for Microorganisms) [78]. A universal data model containing information on the resources (identification, molecular characterisation, physio-chemical characterisation, etc.), history (provenance, acquisition, date of storage, etc.), and practical management (aliquots, storage conditions, methods of cultivation, the person who entered/update the data, etc.) has never been released, but an ISO standard (ISO 21710:2020, "Specification on data management and publication in microbial resource centres"), although scarcely adopted, was developed in 2020 by the Biotechnology Technical Committee (TC 276) to address this matter [79]. Digitisation and updating the information and metadata associated with the preserved microorganisms are, thus, considered one of if not the major limitations of CCs worldwide.

Even regarding this situation, we cannot avoid mentioning the task force carried out by the SUS-MIRRI.IT project, through which great effort is devoted to developing a national open-source interoperable catalogue of microbial resources (itCCC, Italian Culture Collection Catalogue) and stand-alone software for the management and administration of all Italian CC datasets and activities.

Finally, our survey on the management of major Italian microbial CCs analysed their financial situation to ensure long-term sustainability. The financing support of CCs might have different origins (i.e., institutional, governmental, or private). The majority of the microbial CCs in Italy are financially supported by internal funds (86%) or, in other words, by the institution hosting the collection, and this level of funding might vary over the years. Only a small portion of funding (often less than 10%) comes from the services provided to users and stakeholders, especially when they are business oriented, or from participation in projects financed by national governmental or international funding authorities. This means that, often, microbial CCs do not have a wide range of possibilities to spend on or invest in innovating, researching, benchmarking, and, sometimes, even training personnel, with the consequence of impacting the quality of offered services.

An estimate of the average cost for maintaining a collection, excluding personnel and fixed costs of the institute where the collection is hosted, has been calculated in the range of EUR 5,000–15,000 for most Italian CCs (61%). However, especially in the case of running the facilities of big collections or collections maintaining certification systems, this cost can increase up to EUR 30,000 (a situation observed in 23% of major Italian microbial CCs) or be even higher (in 16% of the CCs). These data are not surprising considering that in Europe, the cost of obtaining a quality certification, which can include consulting services and the acquisition of new equipment (e.g., data loggers or power supply units) and the

updating of others already being used (e.g., maintenance and calibration), is about EUR 20,000 plus EUR 3,000 per year for maintenance [80].

From this general overview of management, it is clear that the future outlook for the sustainability of Italian microbial CCs will need, in addition to the implementation of a quality system and the adoption of international guidelines, an increase in access to funding and resources and the enhancement/diversification of services offered to users together with collaboration with external stakeholders. This goal could only be pursued if innovation, competitiveness, and international visibility will be increased. A good starting point to achieve this is the improvement of digitalisation as well as capacity building, two pivotal aspects on which the EU-funded project SUS-MIRRI.IT is focussed.

## 6. Future Directions

Microbial CCs can be promoters of technological and scientific innovation, with a great economic impact, and the developers of high-level skills for the country in which they operate. However, although they present significant opportunities, their future long-term sustainability poses a formidable challenge.

To guarantee that the services offered by microbial CCs are not only reliable but also excellent and, thereby, attract and nurture a robust and dynamic user community dedicated to delivering exemplary scientific research, financial support is imperative. As already stated above, currently, the main financial entry of Italian CCs (but a similar situation is observed worldwide) is dependent on the commitment of universities or research centres to which the collections belong. Consequently, the funding frameworks between collections might greatly vary, according to the "virtuosity" of the institution, affecting substantial developments required to stay at the forefront of technology. This condition can be very critical for those small microbial CCs often preserving rare, autochthonous, and, therefore, precious microorganisms.

The "fundraising" strategy is a key point for the vital management of Italian microbial CCs, and the attraction of investments and search for partnerships, especially with industry, is fundamental. Notably, industries—mainly biotechnological ones—can act as partners in co-designing and co-developing projects to provide market-driven services and trigger the so-called virtuous circle of innovation; access to innovative technologies enables research excellence, which, through knowledge transfer, guides technological development, that, in turn, generates innovation in the industry, thereby improving the technologies available for research. At the same time, for users, this enables taking advantage of specifically dedicated access to cutting-edge facilities, top-quality services, and high-level training for personnel.

In addition to collaboration with the private sector, the connection with international and national funding agencies or public authorities as well as access to territorial funds, collectively contribute—if not ensure— necessary support for the growth and sustainability of microbial CCs. This cooperative synergistic approach aligns with the vision of a financially autonomous sustainable institution capable of upholding and enhancing excellence in both research and service provision.

However, to standardise and optimise resources and expertise within different microbial CCs, especially in Italy where the availability of resources is extremely fragmented, participation in a sovra-structured national or international organisation, such as a research infrastructure, can be extremely strategic and of great value.

Over several decades, Research Infrastructures (RIs) are supposed to catalyse technical excellence, scientific innovation, and research development by coordinating the activities of internal partners and stimulating the exchange of knowledge, proposing synergies and networks with territorial bodies, public institutions, and policy-making authorities; encouraging the aggregation of skills, structures, technologies, and resources including human capital; optimising administrative management; providing IT and ELSI (ethical, legal, and social aspects) support; and, last but not least, promoting future strategic long-term financial planning.

MIRRI-ERIC, the European network of mBRCs, is one of the RIs coordinated by ESFRI, the European Strategy Forum on Research Infrastructures. The establishment of the Italian National Node of this network, MIRRI–IT, in the near future, should act as a revolution in the scenario of Italian microbial biobanks.

MIRRI–IT will be, indeed, an infrastructure distributed throughout the Italian territory, which will include mBRCs capable of providing microorganisms, metadata, and services of outstanding quality according to internationally recognised standards, complying with the Partner Charter of the organisation (i.e., regulating access policy, Nagoya Protocol compliance, data protection, management policy, informed consent, intellectual property rights, biosecurity issues, etc.). In addition, to provide reliable performance that ensures full satisfaction of the needs and requests of the various stakeholders, the Italian MIRRI–IT node will have to guarantee that all mBRCs belonging to the network follow well-defined governance and the management of resources, data, and risk.

Moreover, from the perspective of long-term sustainability, MIRRI–IT will support the Italian mBRCs in the continuous implementation of their facilities with state-of-the-art technology, thereby the visibility and attractiveness among users; it will centralise, through the creation of a common Collaborative Working Environment (CWE), the coordination of all operational activities of the Italian mBRCs, from the interactions with customers to the connection with other RIs and policy decisionmakers; guarantee the commitment of all centres to provide excellent materials and services; and, finally, foster advancements in Italy's green, digital, ecological, and inclusive transition by promoting the circular economy, the development of renewable energy sources, and more sustainable and eco-efficient scientific progress.

## 7. Conclusions

The last few decades have shown how the loss of biodiversity, mainly due to climate change, has a strong impact on the functioning of ecosystems and, thus, on human activities. Microbial CCs and mBRCs make a significant contribution to the protection and valorisation of microbial biodiversity, which is too often underestimated and not taken into consideration when analysing the causes/effects of the global biodiversity impoverishment and the destruction of natural habitats with negative impacts on the planet's health.

Microbial CCs and mBRCs can not only facilitate scientific research by enhancing the conservation, preservation, and characterisation of microbial resources and promoting human and environmental well-being via the industrial biotechnological exploitation of microbial potential, but can also be extremely useful in tackling the major global challenges facing our planet and in achieving the UN Sustainable Development Goals [81,82].

Access to microbial resources indeed plays a crucial role in various sectors of the bioeconomy. These sectors span a wide range of applications—agriculture (biocontrol and biofertilisation), environment (bioremediation of contaminants), food making (starter cultures and fermentation processes), chemicals (bio-catalysis), cultural heritage (microorganisms for the bio-restoration of art works), health (vaccines, antibiotics, and diagnostics), biofuel production, cosmetics, and even forensic and space research [83]. Moreover, the power of the new "pioneering tool" of microbiomes has just started to be explored, and we do not yet know how much microbiomes will impact aspects of our future lives.

Italy possesses an extreme richness and uniqueness of microbial genetic diversity preserved in CCs (e.g., microalgae and viruses, which are poorly represented in other microbial CCs worldwide, and microbiomes from different environments that have already started to be kept), often linked to typical products and excellence characterising the "Made in Italy" label (e.g., dairy products, wine, etc), which could be exploited and valorised in many other fields of global interest. The present review has provided a snapshot of the current situation of Italian microbial CCs, emphasising the potential benefits that unlocking microbial biodiversity preserved in Italy can offer to global society. These benefits include contributing to a more sustainable, competitive, and resilient bioeconomy.

**Author Contributions:** Conceptualisation, G.C.V., M.M. and J.T.; writing—original draft preparation, M.M. and J.T.; writing—review and editing, M.M., J.T., G.P.A., M.S.B., V.B., A.B., M.B.B., M.B., P.B., S.C. (Stefania Carrara), F.C., C.E.C., R.C., S.C. (Sofia Cosentino), A.d., P.D.D., L.G., M.G., S.L., A.M., A.N., G.P., A.M.P., I.P., M.P., A.P. (Annarita Poli), A.P. (Antonino Pollio), A.R. (Anna Reale), A.R. (Annamaria Ricciardi), C.S., L.S. (Laura Selbmann), L.S. (Luca Settanni), S.T., B.T., P.V., M.Z. and G.C.V.; supervision, G.C.V.; funding acquisition, G.C.V. All authors have read and agreed to the published version of the manuscript.

**Funding:** This work was granted by the European Commission—NextGenerationEU, Project SUS-MIRRI.IT "Strengthening the MIRRI Italian Research Infrastructure for Sustainable Bioscience and Bioeconomy", code n. IR0000005.

**Institutional Review Board Statement:** Not applicable.

**Informed Consent Statement:** Not applicable.

**Data Availability Statement:** Parts of the original datasets presented in this review are openly available in the SUS-MIRRI.IT website at https://www.sus-mirri.it (accessed on 31 January 2024). The raw data supporting the statements in this review will be made available by the authors on request.

**Conflicts of Interest:** The authors declare no conflicts of interest.

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
