# Peer review of "Treasures of Italian Microbial Culture Collections: An Overview of Preserved Biological Resources, Offered Services and Know-How, and Management"

_sustainability, doi:10.3390/su16093777_

Round 1
Reviewer 1 Report
Comments and Suggestions for Authors
This review presents the state-of-the-art of Italian CCs and mBRCs, highlighting strengths, weaknesses, threats, and opportunities. Microbial CCs and mBRCs can not only facilitate scientific research by enhancing the 763 conservation, preservation, and characterisation of microbial resources, or promote hu-764 man and environmental well-being via the industrial biotechnological exploitation of the 765 microbial potential but can also be extremely useful in tackling the major global challenges 766 facing our planet and in achieving the UN Sustainable Development Goals.
This review is well organized, which have important implications for the related topics. It can be accepted for publication after minior revision on figures, there are too many bar plots, it is better to change some of the figures.
Author Response
Dear Editor,
We would like to thank the reviewers for the helpful suggestions. We have addressed all concerns, and you can find all replies to the comments below.
Reviewer 1
This review presents the state-of-the-art of Italian CCs and mBRCs, highlighting strengths, weaknesses, threats, and opportunities. Microbial CCs and mBRCs can not only facilitate scientific research by enhancing the conservation, preservation, and characterisation of microbial resources, or promote human and environmental well-being via the industrial biotechnological exploitation of the microbial potential but can also be extremely useful in tackling the major global challenges facing our planet and in achieving the UN Sustainable Development Goals.
This review is well organized, which have important implications for the related topics. It can be accepted for publication after minor revision on figures, there are too many bar plots, it is better to change some of the figures.
We thank the Reviewer’s suggestions. We have decided to modify the two bar plots of Figures 2 and 3 to better display data.
Reviewer 2 Report
Comments and Suggestions for Authors
this review "The treasure of Italian microbial culture collections:..." focuses on the state of microbial collections in Italy, achievements and challenges. A description of microbial collections, their samples and conditions for storage of biological material is given. The history of the organization of microbial collections in Italy and the features of their interaction with each other and with clients who donate cultures or receive them for research are shown.
In general, the review is of significant interest from the point of view of familiarity with the peculiarities of the collection’s functioning and the conditions for its interaction with clients. Overall, the review fits the profile of the Sustainability journal and is eligible for publication.
lines 384-392, In my opinion, what was missing in this review was information on the identification of closely related strains within genera, for example, for Rhodococcus, Bacilli, Streptomycetes, for which identification by the 16 S gene and biochemical tests do not allow unambiguous identification of strains.
Author Response
Dear Editor,
We would like to thank the reviewers for the helpful suggestions. We have addressed all concerns, and you can find all replies to the comments below.
Reviewer 2
This review "The treasure of Italian microbial culture collections:..." focuses on the state of microbial collections in Italy, achievements and challenges. A description of microbial collections, their samples and conditions for storage of biological material is given. The history of the organization of microbial collections in Italy and the features of their interaction with each other and with clients who donate cultures or receive them for research are shown.
In general, the review is of significant interest from the point of view of familiarity with the peculiarities of the collection’s functioning and the conditions for its interaction with clients. Overall, the review fits the profile of the Sustainability journal and is eligible for publication.
lines 384-392, In my opinion, what was missing in this review was information on the identification of closely related strains within genera, for example, for Rhodococcus, Bacilli, Streptomycetes, for which identification by the 16 S gene and biochemical tests do not allow unambiguous identification of strains.
We thank the Reviewer’s suggestions. We have added a sentence (lines 389-394) completing the bacterial molecular identification and included 2 additional references.
Reviewer 3 Report
Comments and Suggestions for Authors
Dear authors,
The manuscript The treasure of Italian microbial Culture Collections: an overview of preserved biological resources, offered services and know-how, and management" present multiple very important data on a vital subject. In the current context of a better management of microbial resources and their products in agronomy, horticulture, natural ecosystems, the approach of the authors is welcomed.
The idea of using an entire country as a state-of-the-art manuscript is very good, and this concept should be replicated in other countries too. It will increase both the knowledge on Culture Collections and microbial Biological Resource Centre, and how they will offer know-how and resources for future use of microorganism and applications designed for microbiomes.
Section 2. An Overview of Italian Microbial Culture Collections - cite Table 1 in the text. Try to explain Table 1 and Figure 1 based on their Location / Working field(s) / Main Taxa Preserved. Figures 2 and 3 present this information, but their text is very short. You can use the text from these two figures to complete the text for Table 1 and Figure 2. It will make a condensed form of the entire Results section text.
The Discussion for this section is good and interesting. It makes a connection with international initiatives.
Section 3. The Joint Research Unit MIRRI-IT and the Project SUS-MIRRI.IT
This section is very well organized, and gives in details the information regarding the WP and interrelations regarding Italian microflora status and potential use.
Try to rewrite the paragraph from 306-313. In this form "Data presented in this manuscript" it looks that data are valid for others sections too. You refer only to the data used in this section. Please clarify.
Section 4. Microbial Resources and Service Supply
This section offers an in depth assessment of microbial resources available in Italian Culture Collections, also their category - species / higher taxonomy ranks.
This part offers a methodological description of the entire processes used in identification and culturing of microorganisms. Te information is important for the future development of large interconnected databases.
Section 5. Management of Italian Microbial Culture Collections
This section present the challenges that microbial collections faces in the current and next periods, how that management is done and how it can be improved.
The last section - 6. Future Directions - is a good Discussion section, that point the future development of microbial collections, their promotion and involvement in technological-scientific innovation.
The entire manuscript is very interesting and very useful for microbiology field, both applied and general one, and all the conclusion state the importance of access to microbial resources, the development of microbial collections and the development of different bioeconomy areas.
Author Response
Dear Editor,
We would like to thank the reviewers for the helpful suggestions. We have addressed all concerns, and you can find all replies to the comments below.
Reviewer 3
Dear authors,
The manuscript The treasure of Italian microbial Culture Collections: an overview of preserved biological resources, offered services and know-how, and management" present multiple very important data on a vital subject. In the current context of a better management of microbial resources and their products in agronomy, horticulture, natural ecosystems, the approach of the authors is welcomed.
The idea of using an entire country as a state-of-the-art manuscript is very good, and this concept should be replicated in other countries too. It will increase both the knowledge on Culture Collections and microbial Biological Resource Centre, and how they will offer know-how and resources for future use of microorganism and applications designed for microbiomes.
Section 2. An Overview of Italian Microbial Culture Collections - cite Table 1 in the text.
We thank the Reviewer’s suggestions. In line 152, Table 1 is cited.
Try to explain Table 1 and Figure 1 based on their Location / Working field(s) / Main Taxa Preserved. Figures 2 and 3 present this information, but their text is very short. You can use the text from these two figures to complete the text for Table 1 and Figure 2. It will make a condensed form of the entire Results section text.
We have added a paragraph in Section 2 highlighting the main characteristics of Italian CCs. Then we have included in the text a better description of Fig. 2.
The Discussion for this section is good and interesting. It makes a connection with international initiatives.
Section 3. The Joint Research Unit MIRRI-IT and the Project SUS-MIRRI.IT
This section is very well organized, and gives in details the information regarding the WP and interrelations regarding Italian microflora status and potential use.
Try to rewrite the paragraph from 306-313. In this form "Data presented in this manuscript" it looks that data are valid for others sections too. You refer only to the data used in this section. Please clarify.
We have slightly changed the paragraph. However, as reported, all manuscript data are, indeed, also valid for other sections and not only for this one.
Section 4. Microbial Resources and Service Supply
This section offers an in depth assessment of microbial resources available in Italian Culture Collections, also their category - species / higher taxonomy ranks.
This part offers a methodological description of the entire processes used in identification and culturing of microorganisms. The information is important for the future development of large interconnected databases.
Section 5. Management of Italian Microbial Culture Collections
This section present the challenges that microbial collections faces in the current and next periods, how that management is done and how it can be improved.
The last section - 6. Future Directions - is a good Discussion section, that point the future development of microbial collections, their promotion and involvement in technological-scientific innovation.
The entire manuscript is very interesting and very useful for microbiology field, both applied and general one, and all the conclusion state the importance of access to microbial resources, the development of microbial collections and the development of different bioeconomy areas.